# Optimization of Fresh Storage of Pistachio (*Pistacia vera* L.) by Use of Different Coatings Under Vacuum

**DOI:** 10.3390/foods14203533

**Published:** 2025-10-17

**Authors:** Ahmet Şahan, Hüseyin Bozkurt

**Affiliations:** 1Pistachio Research Institute, University Avenue No. 34-1, 27060 Gaziantep, Türkiye; ahmet.sahan@tarimorman.gov.tr; 2Department of Food Engineering, Faculty of Engineering, Gaziantep University, 27310 Gaziantep, Türkiye

**Keywords:** fresh pistachio, *Aspergillus flavus*, storage, ozone, chitosan, alginate, zein

## Abstract

The aim of this research was to extend the freshness of pistachio (*Pistacia vera* L.) by using edible coatings during cold storage. Different coatings—chitosan with potassium sorbate, alginate with propionic acid, and zein with EDTA—were used for both in-hull and dehulled pistachios. Effects of coatings on *Aspergillus flavus* count, peroxide, free fatty acids (FFA), water activity, and aflatoxin levels were investigated. Principal component analysis (PCA), correlation analysis, and multi-objective optimization were applied to interpret the data. The used coatings and pistachio hulls were effective in slowing down the formation of FFA and peroxides (*p* < 0.05). Zein and chitosan coatings prevented *Aspergillus flavus* growth up to 5 months while the alginate coating provided superior sensory preservation. PCA revealed that pistachio hulls decreased the aflatoxin, FFA, and peroxide formation, and the hulls of the pistachios showed a protective effect. Also, sensory parameters had negative correlations with FFA and peroxide value. From the optimization study, the best way to increase the freshness of pistachio is through the use of an alginate coating for in-hull. This study showed that alginate and chitosan coatings combined with ozone treatment and vacuum packaging can be used to prolong the freshness of pistachio.

## 1. Introduction

Pistachio, *Pistacia vera* L., is one of the most important agricultural and economic products for Türkiye, which ranks third in world after USA and Iran production [1]. Pistachio harvest is carried out from the beginning of July to the end of September, depending on the variety and climate conditions. Therefore, fresh pistachio consumption can only be performed between these dates as the shelf life of fresh pistachios is very short and is limited to a few days even in cold storage. For this reason, pistachios cannot be consumed as fresh in other places. Fresh pistachios, which have a unique flavor compared to dried or roasted pistachios, are highly appreciated by the people during the harvest season in the provinces where pistachios are harvested. However, since the storage life of fresh pistachios is very short, their trade has not developed. Making fresh pistachios consumable throughout the year by extending their shelf life will create a new market in the domestic and foreign markets and will ensure that people can access this product not only at the end of summer and in a limited region, but at all times.

Edible films and coatings are defined as thin protein-, polysaccharide-, and lipid-based layers formed between food components or on the surface of food in order to prevent quality losses and spoilage reactions in foods, to extend shelf life, and to preserve sensory properties [2]. The use of edible films as packaging materials has attracted attention in recent years because they require simple production technology, are cheap, are obtained from natural compounds, have a variety of functional properties, and are biodegradable [3]. Their functions are to act as a barrier to mass transfer (water, gas, and oils), to act as a carrier of food ingredients and additives (color compounds, flavor compounds, etc.), or to provide mechanical and microbial preservation [4]. Along with all these, edible films and coatings aim to improve the quality of food and preserve its freshness. Edible coatings on fruits and vegetables after harvest are used to preserve color, acids, sugar, taste, and aroma substances, to reduce storage defects, and to obtain a product that appeals to consumer taste.

Ozone has been increasingly used in the food industry in many countries, especially in the last decade. Ozone was accepted as GRAS by the Food and Drug Administration (FDA) in 1982 and is used as a disinfectant in bottled water [5]. The basis for the use of ozone in the food industry is that it is 52% stronger than chlorine. The mechanism of action of ozone on microorganisms is killing by oxidation of cell walls [6,7,8].

Fresh pistachios have a very short shelf life, and the main spoilage mechanism is mold growth, peroxide, and FFA formation, therefore resulting in color (black color), odor, taste, and structural changes. Moreover, aflatoxin production can occur as a result of mold growth. By preventing these spoilage mechanisms in fresh pistachios, the product will be safer and high quality in terms of increasing the physico-chemical quality attributes (such as oxidation).

The aim of this study is to extend the shelf life of vacuum-packed fresh pistachios using edible food coatings with antifungal properties. For that purpose, in-hull and dehulled fresh red pistachios were treated with ozone to remove microbial flora on the raw material, and pure Aspergillus flavus culture was inoculated to clearly observe the effect of coatings. Then, the fresh samples were coated with different edible coating materials and stored to determine the ability of coatings to prevent FFA, peroxide, and aflatoxin formation and *Aspergillus flavus* growth.

## 2. Materials and Methods

### 2.1. Materials

All the solvents used for the aflatoxin standards were purchased from Supelco/Sigma Aldrich ( St Louis, MO., USA). Aflatest immunoaffinity columns were purchased from Vicam Company (Milford, CT., USA). All solvents used for the study were of either analytical or HPLC grade.

Siirt variety pistachios, harvested at 15 September 2022, were used as raw material in the study. The 25 years old trees were selected in the orchard of Republic of Türkiye, Ministry of Food, Agriculture and Livestock, Pistachio Research Institute (Gaziantep, Türkiye). All other materials were analytical grade.

### 2.2. Sample Preparation

The experimental set-up is given in Figure 1. Half of the harvested pistachios were left with their red hulls (in-hull) and the other part were dehulled to obtain hard shelled pistachios by removing the red hulls. As shown in Figure 1, ozone was treated for pasteurization. Two control samples were not treated with ozone.

During ozone application, pistachios were placed in a plastic container, and the lid of the container was closed. A silicone hose was fixed to the 5 mm diameter hole opened in the lid of the container, with the help of cold silicone, in an airtight manner. Then, this hose and the ozone generator hose were combined to ensure that the ozone was transferred to the pistachios homogeneously. After that, the ozone generator was turned on, and the fruits were exposed to ozone for 10 min to pasteurize them.

After pasteurization, the microbial load was determined again to determine the effectiveness of ozone treatment. After this process, pure *Aspergillus flavus* was inoculated into the samples at 3 log (low dose) and 5 log (high dose) log cfu/g. The main goal here was to clearly see the effects of coatings on *Aspergillus flavus*, aflatoxin production, and other quality parameters.

### 2.3. Sample Coating

Chitosan coating: The fresh pistachios were coated with a chitosan–potassium sorbate solution as an antifungal agent. Chitosan (Sigma Chemical Co., St louis, MO, USA) was dissolved in distilled water at 3 g per 100 mL of water. Then, glacial acetic acid (Sigma Chemical Co., St louis, MO, USA) was added at 1 mL per 100 mL of solution. After addition of 4 g potassium sorbate (Sigma Chemical Co., St louis, MO, USA) per 100 mL, the solution pH was adjusted to 5.6 with 1M NaOH (Sigma Chemical Co., St louis, MO, USA), and then 0.5% Tween-80 was added as a plasticizer [9]. Dipping of the fresh pistachios into the solution was carried out for 1 min, and then fruits were left to dry for two hours.

Zein Coating: Firstly, 6.75 g of zein (Sigma Chemical Co., St louis, MO, USA) was dissolved in 40.6 mL of 95% ethanol (Sigma Chemical Co., St louis, MO, USA), and then 1.9 mL of glycerin was added as a plasticizer. The solution was boiled for 5 min, and 5.58 mg/mL EDTA (Sigma Chemical Co.,St louis, MO, USA) was added. The pistachios were dipped for 1 min in this solution. Then, the pistachios were dried at room temperature for 2 h [10].

Alginate Coating: Alginate (Sigma Chemical Co., St louis, MO, USA) was dissolved in pure water at 1% (*w*/*v*) by heating at 45 °C. Then, the solution was brought to room temperature and glycerol (2% *v*/*v*) was added [11]. After that, 0.5% (*w*/*w*) propionic acid (Sigma Chemical Co., St louis, MO, USA) was used as the active substance. The application was carried out in the form of dipping for 1 min, and the fruits were left to dry for two hours after dipping.

### 2.4. Vacuum Packaging and Storage

The packaging of fresh pistachios with different applications was made using a vacuum machine. Each group of pistachio samples, 1000 g, was placed in 15 packs (5 storage times × 3 replicates) consisting of polyamide/polyethylene for vacuum packaging. The vacuum-packed fresh pistachio samples were stored at 5 °C for 5 months. A total of 120 samples (2 different samples as red dehulled and in-hull, 3 coating type and 1 control, 5 storage time and 3 replications) were packed. During the storage period, FFA content, peroxide value, total mold count, and aflatoxin level of the samples were analyzed. After packaging of the samples, they were stored at 5 °C and analyzed. The dates for the analyses were; harvesting (September 15), October 15, November 15, December 15 and January 15, and February 15.

### 2.5. Water Activity Measurements

The water activity value was measured for all samples monthly during storage by water activity measurement set. (Aqualab 4TE). The samples were placed in a sealed steel chamber, and the humidity of the product in the chamber and the air was expected to come into balance. The equilibrium humidity value reached at the end of the process was measured directly with the help of a probe placed in this chamber.

### 2.6. Total Mold Count

The mold count was carried out using a method described in a study by Tournas et al. [12]. About 50 g of sample was taken under aseptic conditions and serially diluted up to 10^−3^ with 450 mL of 0.1% peptone water. Under aseptic conditions, 0.1 mL diluted samples were inoculated on Dichloran Rose Bengal Chloramphenicol (DRBC) agar (Sigma Chemical Co., St louis, MO, USA) in three replicates by the spread-plate method. Incubation was carried out in the dark at 25 °C for 5 days. After the incubation, colonies were counted and expressed as colony forming units (cfu) and calculated as log cfu/g.

### 2.7. Aflatoxin Analyses

Aflatoxin analysis was carried out by use of HPLC (Agilent 1200 Series, Santa Clara, CA, USA) according to the AOAC [13] method. About 1 kg of pistachio sample was ground and homogenized in a blender (Waring, Stamford, CT, USA), and 50 g of ground sample was taken; 5 g of NaCl and 250 mL of extraction solution (methanol:water 8:2) were added into the blender bowl and mixed at high speed for 3 min. The extract was filtered through filter paper (Whatmann No:4). About 20 mL of the filtrate was taken, and 20 mL of ultrapure water was added and then filtered again using a microfiber (Whatmann 1.6 μm glass microfiber). About 10 mL of the filtrate was passed through the immuno affinity column at a rate of 2–3 mL/min and then the column was washed with 10 mL of distilled water. Finally, 1 mL of methanol was passed through the column at a rate of 1–2 drops/second, and the elute was taken into the vial. Again, 2–3 mL of air was passed to ensure that no methanol remains in the column. Then, 1 mL of ultrapure water was passed through the column, taken into a vial, and mixed in a tube mixer. An amount of 1.8 mL of the filtrate was taken into a small vial, and the injection stage was started. The filtrate was placed in auto-sampler and read in HPLC for 15 min after 100 µL injection.

HPLC Terms

Mobile phase = Water: Methanol: Acetonitrile (5:3:2)

HPLC column = C-18 (250 mm × 4.6 mm id)

Flow rate = 1 mL/min

Detector = Fluorescence (Ex: 360, Em: 440)

Multiplication factor (multiplier) = 2

Derivatization = Cobra Cell unit was used to differentiate the aflatoxin peaks.

### 2.8. Determination of FFA (%)

The FFA determination was made according to Shahidi and Zhong [14]. Firstly, 1.00 ± 0.01 g of the oil sample obtained by cold extraction was weighed and dissolved in 75 mL of 95% ethanol. After addition of 3–4 drops of phenolphthalein, the sample was titrated with 0.01 N KOH dissolved in ethanol until the color turned to pink. The result was calculated as FFA (%) (as oleic acid) using the formula below.

FFA (as oleic acid), % = V × N × 28.2/M

N: Normality of KOH solution

V: Volume of spent KOH solution, mL

M: Sample quantity, g 28.2: 282 (molecular weight of oleic acid) × 100/ 1000

### 2.9. Determination of Peroxide Value

The peroxide value (PV) of the pistachio oils was measured according to AOCS Official Method [15]. Pistachio oil obtained from cold extraction was weighed with a precision of 1.00 ± 0.01 g, taken into a 250 mL flask; 30 mL of acetic acid–chloroform mixture (3:2 *v*/*v*) was added, and the oil was dissolved. Then, 0.5 mL of saturated KI was added and shaken continuously and at high speed for 1 min. Then, 30 mL of distilled water was added without waiting and titrated with 0.01 N sodium thiosulfate solution. The result was calculated using the formula below.

Peroxide number value (meq O_2_/kg oil) = (S−B) *×* N *×* 1000/M

Q: Volume of solution spent in titration, mL

B: Volume of solution spent for blank in titration, mL

N: Normality of sodium thiosulfate solution

M: Sample quantity, g

### 2.10. Determination of Moisture Content

About 100 g of pistachio was weighed into a porcelain dish and put into an oven at 105 °C [13]. After constant weigh was reached, the sample was weighed again, and the percent moisture content was calculated as:
$$\%\;\mathrm{Moisture}\;\mathrm{Content}\;=\frac{W1\;-W2}{W1}\times 100$$ where W1 is the amount of sample before drying and W2 is the amount of sample after drying. The dried pistachio samples were used for further analyses.

### 2.11. Sensory Analysis

Sensory analyses were carried out with fresh pistachio samples in a well-aerated room with 10 trained people (5 males and 5 females), who were food and agricultural engineers employed in the Pistachio Research Institute. They were chosen from non-smokers and highly odor-sensitive engineers. Fresh pistachio samples were given to panelists, and they were asked to score samples in terms of taste, odor, texture, and general acceptance.

### 2.12. Statistical Analyses

All analyses at each sampling time were carried out in three replications and results were reported as mean ± standard deviation. The results were compared with analyses of variance (one-way and two-way) and Duncan’s multiple range test using SPSS 13 software at α = 0.05 level [16].

Optimization analysis was carried out to only find the best coating type (chitosan, alginate, or zein) and hull condition of pistachios (storage as in-hull or dehulled) in Design Expert 13 trial version. To find the best conditions, FFA contents, peroxide value total mold count, aflatoxin B1, and total aflatoxin values were adjusted as minimized, and taste, odor, texture, overall acceptability, Aw, and moisture content values were adjusted as maximized in the program.

Correlation analyses were also carried out to find the relation between features of the pistachio samples using SPSS 13 software and the Pearson correlation coefficient was taken into account.

Principal component analysis (PCA) was utilized to interpret relations between different parameters of dehulled and in-hull pistachio using OriginPro 2023 (OriginLab Corp., North Hampton, MA, USA) software.

## 3. Results and Discussion

### 3.1. FFA

Table 1 shows the change in the FFA contents of the fresh pistachio samples during the storage period. FFA values increased over time in all samples. This is expected; lipid oxidation and hydrolysis increase with extended storage. FFA (FFA) contents of fresh pistachios ranged between 0.140% and 0.215% at the harvesting time, and it was continuously increased during the storage period as expected. At the end of the storage, the lowest FFA content was 0.477% and the highest was 1.178%. Generally, the FFA content of the dehulled samples was higher than those of the in-hull samples (*p* < 0.05). The reason for this could be due to the protective effect of the pistachio hull. Moreover, the FFA content of the control samples was higher than those of the coated samples. This showed that the edible coatings can show protective effect on the formation of FFA. Although the coatings slowed the FFA increase, the protective effect in dehulled samples was limited. Also in the control groups, there was a sharp increase starting from month 4. The best coating for the prevention of FFA formation was found to be alginate coating for in-hull pistachios, as they had the lowest (*p* < 0.05) FFA for high doses and the lowest FFA together with the chitosan coating for low doses. The best coating for the prevention of FFA formation was found to be alginate coating for in-hull pistachios, because it is well-known that alginate decreases unsaturated fatty acids oxidation [17], so the alginate-coated pistachios had the lowest (*p* < 0.05) FFA.

As can be seen in Table 1, alginate seemed to be the best coating for the high-dose application in the dehulled samples, while chitosan coatings were the best for low-dose application of the dehulled samples. The highest FFA content was detected at the control samples of the dehulled pistachios for both low and high doses. Hadorn et al. [18] reported that if the FFA value exceeds 1%, pistachio products are considered spoiled. In our study, fresh pistachio samples were considered inedible in terms of FFA starting after the fourth month as the FFA content of the high dose dehulled samples were starting to exceed 1%. Moreover, because FFA value higher than 0.7% is an indicator of rancidity [19] so it was thought that in-hull fresh pistachios could be stored for 5 months without lipid oxidation. However, the shelf life of dehulled fresh pistachios was shorter, almost 2 months according to [20]. Öztürk et al. [21] stored fresh pistachio with different packages for 1 month. They determined that the FFA content of fresh pistachio increased from 0.36% to 0.86% and the lowest increase was found in vacuum packed as from 0.36% to 0.65%. In our study, the initial FFA and the increase in FFA content during storage was lower than those found in Öztürk et al.’s [21] study. The reason for this could be due to the fact that the pistachio samples were not bought from the market, but they were harvested from the trees and treated for coating on the same day. Molamohammadi et al. [22] coated fresh in-hull pistachio fruit with salicylic acid-incorporated chitosan, then packaged and stored them for 28 days. They determined that the FFA content of the pistachios was more than 1% after 28 days of storage. It could be concluded in this study that different applications and vacuum packaging could prevent the formation of FFA.

### 3.2. Peroxide Values

The changes in the peroxide values of fresh pistachios were observed for 5 months of storage and the results are given in Table 2. As seen in Table 2, a significant increase occurred in peroxide values as expected, with increasing storage time which was the indication of increase in oxidative rancidity. Peroxide values were increased sharply after 4 months of storage. The peroxide values of fresh pistachio samples ranged between 0.132 meq O_2_/kg and 0.431 meq O_2_/kg at the beginning of the storage, and these values reached to a level between 0.798 meq O_2_/kg and 1.796 meq O_2_/kg as they were expected. A significant difference was observed between in-hull and dehulled samples (*p* < 0.05), where the peroxide values of in-hull samples were changed between 0.798 meq O_2_/kg and 1.293 meq O_2_/kg, while that of the dehulled samples were in the range of 1.493 meq O_2_/kg to 1.796 meq O_2_/kg. This difference could be due to the protective effect of the red hull on diffusion of oxygen into the pistachios. In all samples, the peroxide values of the control samples were higher than those of the coated samples. It was thought that the coating materials formed a barrier on the pistachio samples, which prevented oxygen diffusion into pistachios. Although the differences between coating materials were not found to be significant (*p* > 0.05) in terms of peroxide value after 5 months, the lowest peroxide value was observed in the zein coating of low-dose in-hull pistachios (*p* < 0.05) and in the zein-coated samples of the high-dose dehulled pistachios (*p* < 0.05). However, at the end of the storage period (after 5 months), the lowest peroxide values were observed in coated in-hull pistachios (*p* < 0.05). Although the differences between coating materials were not found significant (*p* > 0.05) in terms of peroxide value, the alginate and the zein coating were the most effective in retarding peroxide formation. The lowest peroxide value was observed in the low dose of zein-coated in-hull pistachios (*p* < 0.05) and in the high dose of zein-coated samples of the de-hulled pistachios (*p* < 0.05). Molamohammadi et al. [22] found that the peroxide value of pistachios increased over 2.5 meq O_2_/kg after 28 days of storage. In this study, the peroxide values of the pistachios did not exceed 1.796 meq O_2_/kg due to the fact that the pistachios were vacuum packaged, which prevented the peroxide formation. Also, Kazemi et al. [23] studied the effects of the modified packaging by spraying nano ZnO to extend the shelf life of fresh pistachio. They observed that the peroxide values of modified packed and control group were 0.52 meq O_2_/kg and 0.7 meq O_2_/kg after 3 months of storage, respectively. In this study, the peroxide values were found to be similar to the results observed by Kazemi et al. [23] for 3 months storage.

### 3.3. Total Mold Count Results

The total mold count was carried out during storage because microbiological activity is important in terms of food spoilage and chemical changes, and molds in particular cause aflatoxin formation in pistachios. Molds are found everywhere in nature as well as on pistachio hulls, and the pasteurization could inhibit the mold. Ozone was not applied to the control samples, so mold activity was generally observed in these samples. After the pasteurization, pure *Aspergillus flavus* was inoculated and then coated with edible films to see the effectiveness of the coating materials to prevent *Aspergillus flavus* activity.

The total mold counts are given in Table 3 as log cfu/g. As can be seen in Table 3, at the beginning of the storage, mold activity was mainly detected in control samples. At the beginning, all coating materials prevented *Aspergillus flavus* activity. During the 5 months of the storage, mold activity was not observed in all samples of chitosan- and zein-coated pistachios, although high dose, and about 5 log cfu/g of *Aspergillus flavus* was inoculated. This showed the ability of these two coating materials to prevent mold activity in fresh pistachios with incorporation of antifungal substances. The success of the chitosan can be explained by its antimicrobial properties which disrupt the microbial cell membrane with its positively charged structure.

However, during the storage, mold activity was observed on alginate-coated samples of both in-hull and high dose of dehulled pistachios. So, it could be said that alginate is not as good as chitosan and zein in terms of preventing mold activity in fresh pistachios. Also, the mold activity in the in-hull and dehulled pistachios were significantly different (*p* < 0.05). At the end of the 5 months, total mold count in the dehulled pistachios was higher than those of the in-hull pistachios. It was thought that the reason for this was that the hulls of the fresh pistachios have antimicrobial activity [24], and the hulls of the fresh pistachios act as a barrier and protect the inside of the pistachios against mold activity. The mold activity of the control and the alginate-coated samples increased steadily during the storage period and reached a maximum at the end of fifth month. However, mold growth in alginate-coated samples was lower than the control samples, and also alginate coating prevented the mold growth in low dose samples of dehulled pistachios. Öztürk et al. [21] observed total mold count of vacuum packaged in-hull pistachios as 2.8 log cfu/g at the end of 1 month storage. In this study, similar results were observed as was found by Öztürk et al. [21]. Molamohammadi et al. [22] determined the total mold and yeast of salicylic acid-incorporated chitosan-coated fresh pistachios as 2.66 log cfu/g, which were also similar to results of the current study. Also, Motelica et al. [25] obtained a novel antibacterial packaging, based on alginate as biodegradable polymer, and the obtained films had a good antibacterial coverage, being efficient against several pathogens.

### 3.4. Aflatoxin Results

As aflatoxin formation is a major problem in pistachio processing, the levels of aflatoxin in fresh pistachio samples were also determined during storage. The results of the aflaoxin analyses were given in Table 4. As seen in Table 4, aflatoxin was not detected in all samples at the harvesting time up to fourth month. However, at the fifth months of the storage, aflatoxin formation was observed only in control samples. The measured aflatoxin values were 0.42 and 0.52 (aflatoxin B1 and total aflatoxin) for low dose of in-hull samples and 0.82 and 1.11 for high dose of in-hull samples respectively. The measured aflatoxin values of dehulled samples were lower than that of in-hull samples, which were 0.21 and 0.32 (aflatoxin B1 and total aflatoxin) for low dose and 0.53 and 0.93 for high dose, respectively. The formation of aflatoxin is mainly due to mold growth, which is dependent on temperature and moisture content during storage. As the moisture content of fresh pistachios is very high (between 40–50%), aflatoxin formation is expected during storage period. However, in this study, pistachio samples were stored at 5 °C, which could delay mold growth and therefore the formation of aflatoxin samples. Furthermore, coating materials and antifungal compounds prevented the aflatoxin formation in the coated samples due to the limiting the mold growth. When dehulled and in-hull control pistachios were compared in terms of aflatoxin formation, aflatoxin level in the in-hull pistachios were significantly higher (*p* < 0.05). This was thought to be due to the higher moisture content of the in-hull of fresh pistachios. The aflatoxin levels in control samples after 5 months were well below the legal limits in EU countries, which were 5 ppb for aflatoxin B1 and 10 ppb for total aflatoxin [26]. As a result, aflatoxin formation was prevented in all coated samples, which shows the strong potential of these coating materials to ensure stability and safety of food products during long time storage. Kazemi et al. [23] did not detect aflatoxin in the modified packaging groups, but aflatoxin levels of the traditional packaged fresh pistachios were 2.3 ± 5.2 ng/g for total aflatoxin and 3.1 ± 4.0 ng/g for aflatoxin B1 at the end of 3 months of storage.

### 3.5. Moisture Content and Water Activity

The moisture content of fresh pistachio samples was measured during the storage period, and the results are given in Table 5. As seen in Table 5, moisture contents of in-hull samples were between 45.92 and 46.75 at harvest time, and they decreased steadily to 42.78 and 43.68 at the end of the storage (*p* < 0.05). Also, moisture contents of dehulled samples were between 27.87 and 29.43 at harvest time, and they decreased to 26.15 and 27.80 at the end of the storage (*p* < 0.05). The moisture content of in-hull samples was higher than that of dehulled samples because pistachio hulls have high moisture content. The differences between moisture contents of coating materials in each dose were not significant (*p* > 0.05) at harvest time except high dose of dehulled samples where alginate and zein coated samples had higher moisture (*p* < 0.05) However, at the end of storage, zein-coated samples had the highest moisture content (*p* < 0.05) where zein coating prevented moisture loss more than those of the other coatings. Since zein either in aqueous ethanol or in aqueous acetone forms biodegradable films with a good tensile and water barrier properties [27,28], it was the best for the protection of moisture loss. Moisture loss could occur due to the movement of water toward outside of the fresh pistachios and condensing of water at the outside. Water content is important for fresh pistachios, and loss of water should be prevented for freshness of pistachio during storage. Therefore, pistachio samples were vacuum packaged before storage to prevent water loss and also to prevent mold activity. As a result, water loss from fresh pistachios was minimized. Sheikki et al. [29] reported that the weight loss of fresh pistachios under modified atmospheric packaging and ambient atmosphere varied between 0.5% and 10.9%, respectively, after 105 days of cold storage. Also, Öztürk et al. [21] found that the moisture loss of fresh pistachios was 5.5% under vacuum at cold storage of 1 month. In this study, total moisture loss was changed between 3–9% after 5 months of storage depending on whether the samples were in-hull or dehulled.

The water activities of fresh pistachio samples were also measured during storage period, and the results are given in Table 6. During the storage water activities of samples were slightly decreased depending on decrease in moisture content of pistachios. At harvest time, water activities of samples were changed between 0.957 and 0.972 and the differences between coating materials of each dose were not significant except high dose of dehulled samples where alginate coating had higher water activity (*p* < 0.05). After 5 months of storage, water activity values were decreased to 0.935–0.953 and the differences between storage months were significant (*p* < 0.05). The water activity decreases in control groups were more obvious. The coating materials especially zein and alginate slowed the decline in water activity and preserved the moisture balance of the samples. The water activity of dehulled samples was lower than in-hull samples due to the protective effect of the pistachio hulls against moisture loss. High water activity is the main reason for deterioration of food samples. Therefore, the food samples are dried before storage to decrease water activity to prevent microbiological activity, chemical deteriorations like FFA, and peroxide formation. However, in this study as the pistachios wanted to be stored as fresh, the samples could not be dried and should be stored with their moisture. Therefore, the storage of the pistachios with high water activity was too difficult because the samples were easily degradable due to high moisture content. Different coatings with antifungal agents, ozone application, vacuum packing and low temperature storage was used to prevent deterioration of fresh pistachios due to high water activity. Although the water activities of samples were slightly decreased during 5 months of storage depending on the decrease in moisture content, they were still higher than 0.935 and very sensitive to deteriorations.

### 3.6. Sensory Analysis Results

In sensory analysis, fresh pistachio samples were scored by the panelists in terms of taste, odor, texture and general acceptability and the results are shown in Figure 2. All sensory attributes decreased gradually during storage as expected in all treatments. The highest decline in sensory attributes were in control groups while coated samples preserved sensory quality throughout the storage with respect to control samples. Zein-coated samples preserved textural quality due to its hydrophobic and film forming nature. Sensory attributes were preserved better at in-hull samples due to the protective effect of the hulls.

When taste results were examined the differences between coating materials were not significant (*p* > 0.05) even if alginate took higher scores than the other coating materials.

Similarly, the differences between odor results of coating materials were not significant at the beginning of the storage. However, alginate came to the fore from the third month of the storage in in-hull low-dose samples and also the fourth month of the storage in low- and high-dose of dehulled samples (*p* < 0.05).

Texture was another important parameter for consumer acceptability of fresh pistachios. Since during the storage softening of the pistachios is unwanted satiation and should be slowed down. The differences between coating materials in-hull samples were not significant, although alginate coated samples had higher scores (*p* > 0.05). However, differences between texture scores of dehulled samples were significant (*p* < 0.05) and alginate was come to the fore from the 4 month of the storage.

General acceptability (gen. acc.) scores of the coating materials of in-hull samples were not different until 4 months of the storage (*p* > 0.05). However, alginate-coated samples took higher scores at 4 and 5 months and the differences were significant. In dehulled samples alginate come to the fore with higher scores from the 2 month at low dose and 3 month at high dose samples.

When the scores of the same samples at different storage time were compared, they decreased smoothly and the differences were significant (*p* < 0.05).

The hull of the pistachio did not affect the sensory results because the differences between in-hull and dehulled samples were not significant in terms of taste, odor, texture, and general acceptability (*p* > 0.05). Also, there was no significant difference between low- and high-dose samples.

### 3.7. Correlation Analysis

The results of the correlation analysis were given in Figure 3. Pearson coefficient (r) values below 0.4 were considered as weak correlation, and r values above 0.6 were considered as strong correlation.

There was a strong positive correlation between FFA and peroxide value (r = 0.868 and *p* < 0.01). During the storage, peroxide value increased with increasing FFA content. A positive weak correlation exists between FFA and aflatoxin B1 (r = 0.253, *p* < 0.05) and total aflatoxin values (r = 0.255, *p* < 0.05). Therefore, aflatoxin formation was not directly relevant with FFA formation. Also, there was a very weak positive correlation between FFA and total mold count (r = 0.169, *p* < 0.05). Correlation analysis showed a strong negative relationship between FFA content and sensory quality and also peroxide value and sensory quality, highlighting the importance of this parameter for maintaining the freshness of pistachios. During the storage, increase in FFA content negatively affected taste (r = −0.573 and *p* < 0.01), odor (r = −0.671 and *p* < 0.01), texture (r = −0.696 and *p* < 0.01) and overall acceptability (r = −0.731, *p* < 0.01) of fresh pistachio samples. In most cases, FFA formation contributes to the rancidity because they cause auto-oxidation easier than intact triglycerides and cause off odor and off flavor [30] so negatively affect from the sensory attributes. There were also strong negative correlations between peroxide values and sensory attributes. Increase in peroxide value negatively affected taste (r = −0.727 and *p* < 0.01), odor (r = −0.754 and *p* < 0.01), texture (r = −0.785 and *p* < 0.01) and overall acceptability (r = −0.794 and *p* < 0.01) during storage period. Due to the fact that peroxide formation causes off-odor and off-flavor in foods, as peroxide value increased, sensory properties were adversely affected.

Correlation between peroxide value and aflatoxin B1 (r = 0.289, *p* < 0.01) and peroxide value and total aflatoxin (r = 0.285, *p* < 0.01) is important. However, the correlation was weak so peroxide value did not directly affect aflatoxin level.

Total mold count was positively correlated with aflatoxin B1 (r = 0.250, *p* < 0.05) and total aflatoxin (r = 0.270, *p* < 0.01) but this correlation was not strong. It is known that aflatoxins are produced by molds, especially *Aspergillus flavus,* so a positive correlation between total mold and aflatoxin is expected. However, the presence of molds does not mean the presence of aflatoxins because aflatoxin production requires some suitable conditions such as high moisture and temperature. Therefore, the correlation between aflatoxin and total mold was weak. Also, mold formation negatively affected odor (r = −0.353, *p* < 0.01), texture (r = −0.286, *p* < 0.01) and overall acceptability (r = −0.249, *p* < 0.05) but not the taste (*p* > 0.05) weakly.

Aflatoxin formation was also negatively correlated with the sensory attributes. There was a negative correlation between aflatoxin B1 and taste (r = −0.349, *p* < 0.01), odor (r = −0.349, *p* < 0.01), texture (r = −0.381, *p* < 0.01), and overall acceptability (r = −0.345, *p* < 0.01). Also, total aflatoxin was negatively correlated with taste (r = −0.343, *p* < 0.01), odor (r = −0.333, *p* < 0.01), texture (r = −0.371, *p* < 0.01), and overall acceptability (r = −0.336, *p* < 0.01). However, the correlation was not strong so aflatoxin did not directly affect the sensory attributes.

### 3.8. Optimization

Optimization analysis was carried out to only find the best coating type (chitosan, alginate, or zein) and hull condition of pistachios (storage as in-hull or dehulled) in Design Expert 13 trial version. The best conditions were minimum FFA contents, minimum peroxide value, minimum total mold count, minimum aflatoxin B1, and total aflatoxin values and also maximum sensory attributes values (taste, odor, texture, and overall acceptability) and maximum Aw and moisture content values. Therefore, Design expert program was set up to find best coating type and hull condition which provide these results.

The result of optimization showed that the best conditions for the storage of fresh pistachio were low dose of alginate coated in-hull samples with a desirability value of 0.919. The second-best condition was low dose of chitosan coated in-hull samples with a desirability value of 0.915. The top eight results of the optimization included in-hull samples.

This result demonstrates that the storage of the fresh pistachios was better with hulls in terms of shelf life. It was thought that in-hull pistachios had better storage stability because the hulls of the fresh pistachios act as a natural barrier and protect the inside of the pistachios from mold activity and oxidative degradation. This also verifies that the pistachio hulls protect the fruit against environmental conditions. Also, it was well-known that pistachio hulls had antifungal activity; therefore, the storage of pistachios with hulls helps to prevent fungal activity.

Alginate including propionic acid was the best coating type for the storage of the fresh pistachios. Although alginate coatings showed less antifungal affect, they were more effective at preserving sensory attributes like taste, odor, and general acceptability so they come to the fore in optimization analysis. This may be due to the moisture retention capacity of alginate due to the forming an oxygen impermeable film and preventing flavor loss and oxidative rancidity.

The second-best coating type was the chitosan including potassium sorbate. However, zein coating including EDTA was the last suitable coating for the storage of fresh pistachios. These results coincide with the previous studies about antifungal properties of chitosan and protein-based coatings because of their ability to break down cell membranes of fungi and extract essential nutrients [31].

Multipurpose optimization, which uses desirability features, constructed a strong frame to determine the best combination of applications and low dose inoculated of alginate coated in-hull samples was determined as best combination with highest desirability score of 0.919. This result indicates that there is a favorable balance between microbial reliability and sensory quality. This approach indicates the importance of integration of statistical modelling into food preservation strategies.

### 3.9. PCA Analysis

PCA analysis was performed to determine the effects of edible coatings on fresh pistachio quality and explore inter-variable relationships and assess the influence of key storage factors. The results are given in Figure 4.

It is clearly seen that control groups were collected at the right side of the graph. That indicates the control samples had the highest peroxide, FFA total mold and aflatoxin values and also had the lowest moisture and water activity values. This result shows that control group spoiled faster and more than coated samples. Alginate, chitosan and zein coated samples collected at left side of the graph indicating low peroxide, FFA, total mold, and aflatoxin values and that shows that the coated pistachio samples spoiled slower than the control samples. The vectors at the left side (taste, odor, texture, and overall acceptability) indicates the coated samples had better sensory attributes than the control samples. Therefore, PCA analyses clearly demonstrate that the alginate, chitosan, and zein coatings extend the shelf life of the fresh pistachio samples by delaying oxidative and microbiological spoilage.

Time, peroxide, and FFA vectors located at the positive part of PC1, which demonstrates FFA and peroxide values strongly correlated with time and increased during storage. However, moisture content and water activity vectors positioned opposite side of time vectors indicating a strong negative correlation. That means that moisture content and water activity were decreased during storage. Also taste, odor, texture, and overall acceptability vectors positioned opposite side of time vectors. That demonstrates strong negative correlation of time and sensory attributes, which means sensory quality of pistachios were decreased with time. Mold and yeast were also positively correlated with the aflatoxin level. Since aflatoxins are produced by molds, positive correlation is expected.

FFA and peroxide value had strong positive correlation with each other. In fact, peroxide value is related with FFA formation (free radicals) and they are the indicator of spoilage of foods and both increases with time during storage. The sensory attributes (taste, odor, texture, and overall acceptability) strongly correlated with each other, namely, it shows that the product quality perceived by consumers (overall acceptability) is closely related to taste, flavor, and texture.

The points of the chitosan coating were concentrated on the left side of the graph particularly lower part. That position indicates a positive correlation with sensory attributes and negative correlation with peroxide, FFA, and total molds. This suggests that the chitosan coating maintains the sensory quality and texture of the product, while also preventing spoilage (peroxide and FFA increase) and mold and yeast growth over time.

The points of the alginate coating were concentrated on the left side of the graph, particularly the lower part in a region similar to the chitosan group. It can be said that alginate performed better storage stability than control samples. However, it was slightly more central located compared to chitosan; therefore, the protective effect of the alginate was slightly lower than that of chitosan.

The points of the zein coatings were spread out both left and right. This suggests that the effectiveness of zein coating may be more variable than alginate and chitosan.

PCA analyses indicated that chitosan and alginate coatings had the best storage performance by maintaining product quality and preventing spoilage. However, control samples showed the worst performance in terms of spoilage and mold and yeast growth. While zein coating performed better storage stability than the control samples, the effectiveness of zein coating may be more variable than chitosan and alginate coatings.

## 4. Conclusions

This study confirms that the combination of ozone treatment, edible coating, antifungal agents, and vacuum packaging increase the shelf life of fresh pistachio. These results suggest practical applications for the safe consumption of fresh pistachios over a longer period of time, especially in regions where pistachios are a major agricultural product. This suggested method can be applicable on an industrial scale, preserves the pistachio quality, reduces food safety risks, and has the potential to creating a new sector in the pistachio trade. This study is a valuable example of the usage of edible coatings with antimicrobial agents to preserve food quality and safety. However, this study was conducted with a limited type of coatings and antifungal compounds, one pistachio type and one storage temperature due to short harvest time of pistachio and limited funding constraints. Therefore, future studies are needed to focus on the practices of different pistachio types, longer periods of storage with a wider variety of coating and antifungal compound formulations, longer storage temperatures, and different packaging atmospheres.

## Figures and Tables

**Figure 1 foods-14-03533-f001:**
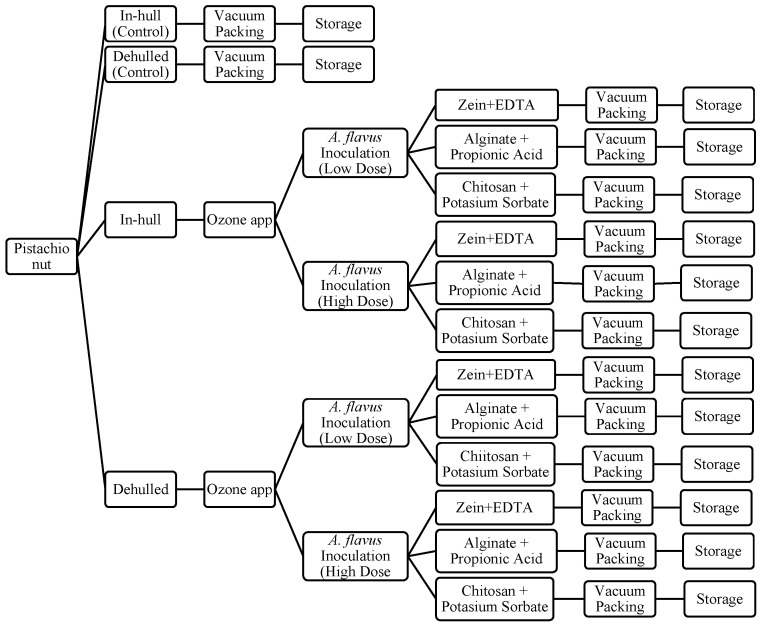
Experimental set-up diagram: sample preparation.

**Figure 2 foods-14-03533-f002:**
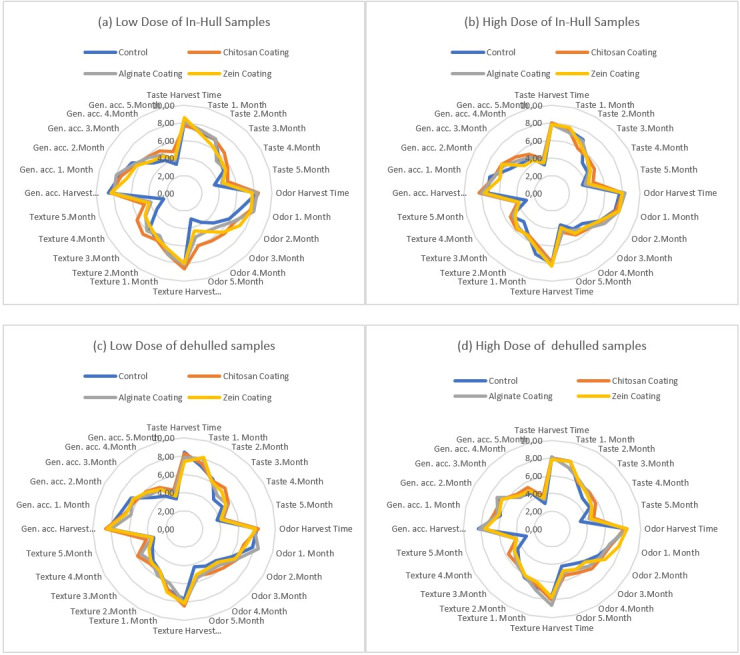
Sensory analysis of (**a**) low dose of in-hull samples; (**b**) high dose of in-hull samples; (**c**) low dose of dehulled samples; and (**d**) high dose of dehulled samples.

**Figure 3 foods-14-03533-f003:**
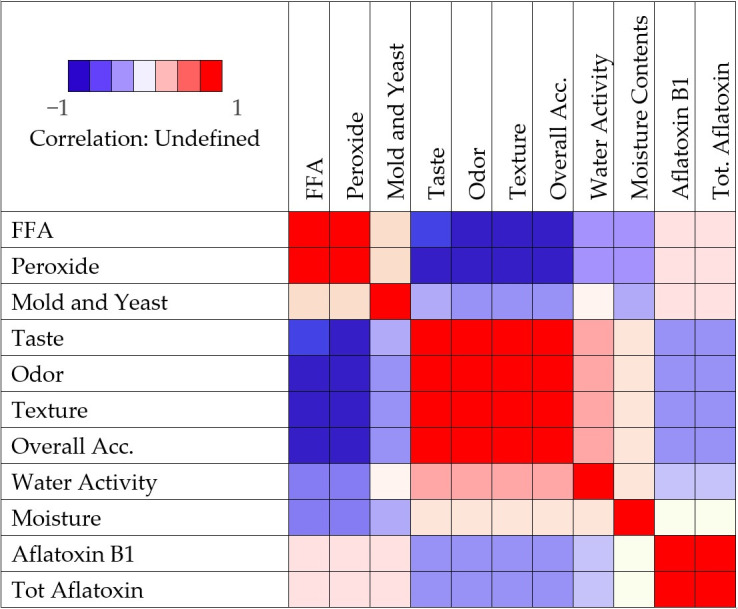
Heat map for some parameters of fresh pistachio during storage.

**Figure 4 foods-14-03533-f004:**
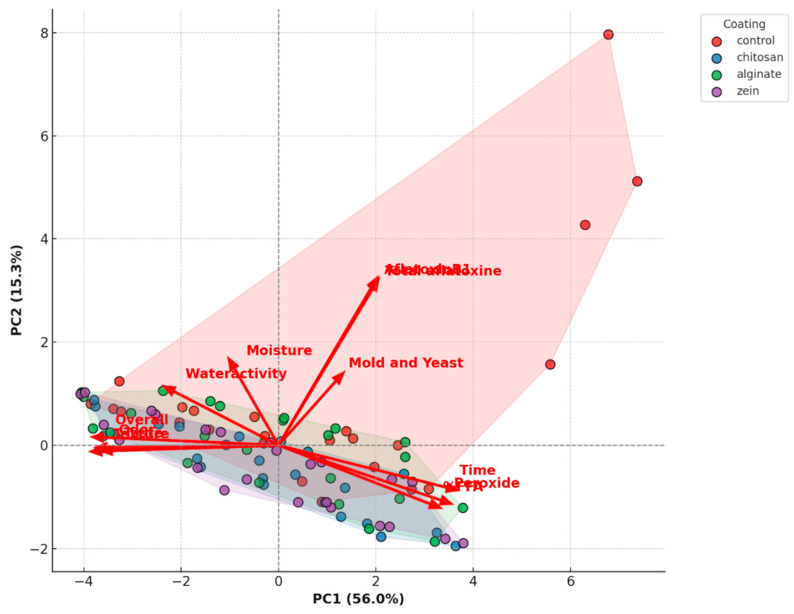
PCA biplot displaying both the coatings (as points) and the variables (as vectors).

**Table 1 foods-14-03533-t001:** FFA Content (%) of pistachio samples during the storage period.

			FFA (% Oleic Acid)					
			Harvest Time	1. Month	2. Month	3. Month	4. Month	5. Month
IN-HULL	Low Dose	Control	0.197 ± 0.01aD	0.327 ± 0.01aC	0.355 ± 0.01aBC	0.365 ± 0.01bBC	0.393 ± 0.016bB	0.700 ± 0.017aA
		Chitosan Coating	0.215 ± 0.01aD	0.215 ± 0.01cD	0.384 ± 0.008aC	0.412 ± 0.01aC	0.478 ± 0.016aB	0.608 ± 0.01bcA
		Alginate Coating	0.196 ± 0.01aD	0.271 ± 0.01bC	0.308 ± 0.01bBC	0.327 ± 0.01cB	0.337 ± 0.015cB	0.561 ± 0.015cA
		Zein Coating	0.150 ± 0.01bE	0.215 ± 0.01cD	0.233 ± 0.01cCD	0.262 ± 0.01dC	0.356 ± 0.01bcB	0.617 ± 0.017bA
	High Dose	Control	0.197 ± 0.01aD	0.365 ± 0.01aC	0.384 ± 0.009aC	0.402 ± 0.015aBC	0.440 ± 0.01aB	0.702 ± 0.016aA
		Chitosan Coating	0.141 ± 0.01bE	0.215 ± 0.01bD	0.238 ± 0.008b CD	0.262 ± 0.009cC	0.355 ± 0.01bB	0.674 ± 0.016aA
		Alginate Coating	0.159 ± 0.01abE	0.224 ± 0.01bD	0.290 ± 0.009cC	0.318 ± 0.009bC	0.393 ± 0.016bB	0.477 ± 0.015cA
		Zein Coating	0.140 ± 0.01bE	0.215 ± 0.01bD	0.262 ± 0.01bcC	0.290 ± 0.009bcC	0.356 ± 0.01bB	0.533 ± 0.016bA
DEHULLED	Low Dose	Control	0.197 ± 0.01abC	0.215 ± 0.01cC	0.224 ± 0.01cC	0.841 ± 0.032bB	0.899 ± 0.015bB	1.133 ± 0.007aA
		Chitosan Coating	0.187 ± 0.02abE	0.272 ± 0.01bcD	0.327 ± 0.01bC	0.729 ± 0.015cB	0.870 ± 0.016bA	0.908 ± 0.01cA
		Alginate Coating	0.150 ± 0.01bF	0.300 ± 0.01bE	0.346 ± 0.01bD	0.599 ± 0.009dC	0.858 ± 0.008bB	0.918 ± 0.01cA
		Zein Coating	0.206 ± 0.01aE	0.833 ± 0.01aD	0.888 ± 0.008aC	0.917 ± 0.009aC	1.000 ± 0.01aB	1.039 ± 0.016bA
	High Dose	Control	0.197 ± 0.01aD	0.299 ± 0.01bC	0.327 ± 0.01aC	0.954 ± 0.015aB	0.994 ± 0.024bB	1.178 ± 0.015aA
		Chitosan Coating	0.187 ± 0.02aD	0.337 ± 0.01aC	0.346 ± 0.01aC	0.936 ± 0.01aB	1.131 ± 0.01aA	1.169 ± 0.01aA
		Alginate Coating	0.215 ± 0.02aE	0.272 ± 0.01cD	0.328 ± 0.01aC	0.674 ± 0.015cB	0.963 ± 0.01bA	1.000 ± 0.01bA
		Zein Coating	0.215 ± 0.01aE	0.300 ± 0.02bD	0.327 ± 0.02aD	0.748 ± 0.024bC	0.972 ± 0.01bB	1.038 ± 0.015bA

Different small letters show significant difference in each dose of each month of storage at α = 0.05 level. Different capital letters show significant difference between the storage months each sample at α = 0.05 level.

**Table 2 foods-14-03533-t002:** Peroxide values (meq O_2_/kg) of pistachio samples during the storage period.

			Peroxide Values (meq O_2_/kg)					
			Harvest Time	1. Month	2. Month	3. Month	4. Month	5. Month
IN-HULL	Low Dose	Control	0.33 ± 0.03aD	0.697 ± 0.05aC	0.764 ± 0.08abC	0.995 ± 0.05aB	1.062 ± 0.03aB	1.227 ± 0.03aA
		Chitosan Coating	0.398 ± 0.05aD	0.597 ± 0.05bC	0.697 ± 0.05aBC	0.797 ± 0.05bB	0.995 ± 0.05aA	1.093 ± 0.05abA
		Alginate Coating	0.133 ± 0.03bE	0.199 ± 0.05cDE	0.332 ± 0.03cD	0.530 ± 0.03dC	0.731 ± 0.03bB	0.995 ± 0.05bA
		Zein Coating	0.099 ± 0.00bC	0.199 ± 0.01cC	0.331 ± 0.03cB	0.730 ± 0.03cA	0.763 ± 0.03bA	0.798 ± 0.05cA
	High Dose	Control	0.431 ± 0.03aD	0.597 ± 0.05aCD	0.697 ± 0.05bC	1.095 ± 0.05aB	1.092 ± 0.05aB	1.293 ± 0.05aA
		Chitosan Coating	0.299 ± 0.05aC	0.698 ± 0.05aB	0.796 ± 0.05aAB	0.827 ± 0.03bAB	0.831 ± 0.05bAB	0.894 ± 0.05bA
		Alginate Coating	0.132 ± 0.03bD	0.233 ± 0.03bCD	0.299 ± 0.01cC	0.629 ± 0.03cB	0.796 ± 0.05bA	0.896 ± 0.05bA
		Zein Coating	0.132 ± 0.03bD	0.199 ± 0.01bCD	0.299 ± 0.01cC	0.298 ± 0.01dC	0.797 ± 0.05bB	0.994 ± 0.04bA
DEHULLED	Low Dose	Control	0.299 ± 0.05aE	0.497 ± 0.00bD	0.729 ± 0.033bC	0.729 ± 0.03cC	1.391 ± 0.057aB	1.796 ± 0.05aA
		Chitosan Coating	0.431 ± 0.01aD	0.597 ± 0.05bC	0.895 ± 0.05abB	0.930 ± 0.05bB	1.394 ± 0.05aA	1.496 ± 0.002cA
		Alginate Coating	0.331 ± 0.03aD	0.496 ± 0.05cC	0.962 ± 0.03aB	1.094 ± 0.05aB	1.461 ± 0.03aA	1.692 ± 0.06abA
		Zein Coating	0.399 ± 0.05aC	0.896 ± 0.05aB	1.030 ± 0.03aB	1.060 ± 0.05aB	1.493 ± 0.05aA	1.793 ± 0.05aA
	High Dose	Control	0.398 ± 0.05aD	0.465 ± 0.03bD	0.794 ± 0.002bC	0.829 ± 0.03bcC	0.998 ± 0.05cB	1.661 ± 0.05aA
		Chitosan Coating	0.399 ± 0.05aE	0.648 ± 0.03aD	0.997 ± 0.05aC	1.094 ± 0.05aC	1.491 ± 0.05aB	1.658 ± 0.03aA
		Alginate Coating	0.364 ± 0.03aE	0.563 ± 0.03abD	0.696 ± 0.05cC	0.963 ± 0.03abB	1.494 ± 0.05aA	1.525 ± 0.03bA
		Zein Coating	0.332 ± 0.03aD	0.697 ± 0.05aC	0.796 ± 0.05bcC	0.796 ± 0.05cC	1.225 ± 0.03bB	1.493 ± 0.05bA

Different small letters show significant difference in each dose of each month of storage at α = 0.05 level. Different capital letters show significant difference between the storage months each sample at α = 0.05 level.

**Table 3 foods-14-03533-t003:** Total mold count (log cfu/g) of pistachio samples during the storage period.

Total Mold Count (log cfu/g)								
			Harvest Time	1. Month	2. Month	3. Month	4. Month	5. Month
IN-HULL	Low Dose	Control	<2	2.22 ± 0.01aB	2.35 ± 0.05bAB	2.46 ± 0.08bA	2.52 ± 0.04bA	2.56 ± 0.04bA
		Chitosan Coating	<2	<2	<2	<2	<2	<2
		Alginate Coating	<2	<2	2.60 ± 0.064aB	2.67 ± 0.03aAB	2.70 ± 0.05aAB	2.73 ± 0.02aA
		Zein Coating	<2	<2	<2	<2	<2	<2
	High Dose	Control	2.15 ± 0.1aC	2.64 ± 0.03aB	2.67 ± 0.05aAB	2.70 ± 0.0aAB	2.75 ± 0.03aAB	2.80 ± 0.02aA
		Chitosan Coating	<2	<2	<2	<2	<2	<2
		Alginate Coating	<2	2.12 ± 0.01bD	2.20 ± 0.01bCD	2.30 ± 0.0bC	2.67 ± 0.032bB	2.75 ± 0.02bA
		Zein Coating	<2	<2	<2	<2	<2	<2
DEHULLED	Low Dose	Control	2.35 ± 0.05aC	3.15 ± 0.017aB	4.85 ± 0.0aA	4.86 ± 0.0aA	4.87 ± 0.0aA	4.90 ± 0.01aA
		Chitosan Coating	<2	<2	<2	<2	<2	<2
		Alginate Coating	<2	<2	<2	<2	<2	<2
		Zein Coating	<2	<2	<2	<2	<2	<2
	High Dose	Control	2.51 ± 0.04aC	4.24 ± 0.0aB	4.27 ± 0.0aAB	4.30 ± 0.01aAB	4.32 ± 0.0aA	4.33 ± 0.01aA
		Chitosan Coating	<2	<2	<2	<2	<2	<2
		Alginate Coating	<2	2.12 ± 0.01bC	2.35 ± 0.05bB	2.41 ± 0.05bB	4.16 ± bA	4.17 ± 0.0bA
		Zein Coating	<2	<2	<2	<2	<2	<2

Different small letters show significant difference in each dose of each month of storage at α = 0.05 level. Different capital letters show significant difference between the storage months each sample at α = 0.05 level.

**Table 4 foods-14-03533-t004:** Aflatoxin analyses results for pistachio samples during the storage period.

			Harvest Time		1. Month		2. Month		3. Month		4. Month		5. Month	
			B1	Total	B1	Total	B1	Total	B1	Total	B1	Total	B1	Total
IN-HULL	Low Dose	Control	nd	nd	nd	nd	nd	nd	nd	nd	nd	nd	0.42 ± 0.0033c	0.52 ± 0.0c
		Chitosan Coating	nd	nd	nd	nd	nd	nd	nd	nd	nd	nd	nd	nd
		Alginate Coating	nd	nd	nd	nd	nd	nd	nd	nd	nd	nd	nd	nd
		Zein Coating	nd	nd	nd	nd	nd	nd	nd	nd	nd	nd	nd	nd
	High Dose	Control	nd	nd	nd	nd	nd	nd	nd	nd	nd	nd	0.82 ± 0.0066a	1.11 ± 0.011a
		Chitosan Coating	nd	nd	nd	nd	nd	nd	nd	nd	nd	nd	nd	nd
		Alginate Coating	nd	nd	nd	nd	nd	nd	nd	nd	nd	nd	nd	nd
		Zein Coating	nd	nd	nd	nd	nd	nd	nd	nd	nd	nd	nd	nd
DEHULLED	Low Dose	Control	nd	nd	nd	nd	nd	nd	nd	nd	nd	nd	0.21 ± 0.0066d	0.32 ± 0.006d
		Chitosan Coating	nd	nd	nd	nd	nd	nd	nd	nd	nd	nd	nd	nd
		Alginate Coating	nd	nd	nd	nd	nd	nd	nd	nd	nd	nd	nd	nd
		Zein Coating	nd	nd	nd	nd	nd	nd	nd	nd	nd	nd	nd	nd
	High Dose	Control	nd	nd	nd	nd	nd	nd	nd	nd	nd	nd	0.53 ± 0.0067b	0.93 ± 0.0066b
		Chitosan Coating	nd	nd	nd	nd	nd	nd	nd	nd	nd	nd	nd	nd
		Alginate Coating	nd	nd	nd	nd	nd	nd	nd	nd	nd	nd	nd	nd
		Zein Coating	nd	nd	nd	nd	nd	nd	nd	nd	nd	nd	nd	nd

Different letters show significant difference at α = 0.05 level.

**Table 5 foods-14-03533-t005:** Moisture content of pistachio samples during the storage period.

		Moisture Content (%)						
			Harvest Time	1. Month	2. Month	3. Month	4. Month	5. Month
IN-HULL	Low Dose	Control	46.75 ± 0.23aA	46.40 ± 0.30aA	44.66 ± 0.15bB	43.62 ± 0.11cC	43.16 ± 0.06bC	42.78 ± 0.10cD
		Chitosan Coating	46.31 ± 0.27aA	46.63 ± 0.21aA	44.64 ± 0.10bB	44.09 ± 0.08bC	43.77 ± 0.05aC	43.15 ± 0.05bD
		Alginate Coating	45.92 ± 0.12aA	46.06 ± 0.08aA	44.37 ± 0.60bB	43.60 ± 0.18cC	43.29 ± 0.20bC	42.51 ± 0.08dD
		Zein Coating	46.08 ± 0.04aA	45.92 ± 0.09aA	45.02 ± 0.51aB	44.62 ± 0.07aC	43.94 ± 0.02aD	43.56 ± 0.02aE
	High Dose	Control	46.49 ± 0.02aA	45.94 ± 0.09aB	44.96 ± 0.09aC	44.63 ± 0.05aD	43.91 ± 0.01aE	43.52 ± 0.05bF
		Chitosan Coating	46.47 ± 0.04aA	45.90 ± 002aB	45.01 ± 0.07aC	44.44 ± 0.01cD	43.88 ± 0.02aE	43.54 ± 0.03bF
		Alginate Coating	46.47 ± 0.03aA	45.90 ± 0.03aB	45.08 ± 0.06aC	44.39 ± 0.01cD	43.93 ± 0.02aE	43.56 ± 0.02abF
		Zein Coating	46.48 ± 0.03aA	45.86 ± 0.01aB	45.01 ± 0.91aC	44.55 ± 0.03bD	43.99 ± 0.05aE	43.68 ± 0.01aF
DEHULLED	Low Dose	Control	29.21 ± 0.25aA	28.40 ± 0.13bcB	28.06 ± 0.07bBC	27.91 ± 0.02abC	27.33 ± 0.12bD	26.77 ± 0.05bE
		Chitosan Coating	29.43 ± 0.15aA	28.72 ± 0.09aB	28.07 ± 0.08bC	27.75 ± 0.11bCD	27.37 ± 0.06bD	26.94 ± 0.20bE
		Alginate Coating	29.05 ± 0.11aA	28.26 ± 0.08cB	27.74 ± 0.11bC	26.91 ± 0.17cD	26.69 ± 0.14cD	26.15 ± 0.04cE
		Zein Coating	28.86 ± 0.03aA	28.62 ± 0.03abB	28.38 ± 0.03aC	28.13 ± 0.01aD	27.91 ± 0.03aE	27.77 ± 0.05aF
	High Dose	Control	28.81 ± 0.08bA	28.53 ± 0.03bB	28.37 ± 0.02aC	28.27 ± 0.04aC	27.93 ± 0.02aD	27.68 ± 0.05aE
		Chitosan Coating	28.93 ± 0.02bA	28.72 ± 0.06aB	28.31 ± 0.10aC	28.02 ± 0.06bD	27.53 ± 0.03aE	27.40 ± 0.04bE
		Alginate Coating	29.21 ± 0.05aA	28.88 ± 0.05aAB	28.55 ± 0.44aBC	28.31 ± 0.05aC	27.77 ± 0.26aD	27.65 ± 0.08aD
		Zein Coating	27.87 ± 0.05bA	28.54 ± 0.05bB	28.36 ± 0.01aC	28.10 ± 0.02bD	27.86 ± 0.05aE	27.80 ± 0.08aE

Different small letters show significant difference in each dose of each month of storage at α = 0.05 level. Different capital letters show significant difference between the storage months each sample at α = 0.05 level.

**Table 6 foods-14-03533-t006:** Water activity of pistachio samples during the storage period.

		Aw						
	Control		Harvest Time	1. Month	2. Month	3. Month	4. Month	5. Month
IN-HULL	Low Dose	Control	0.962 ± 0.01aA	0.95 ± 0.01abAB	0.956 ± 0.01aB	0.950 ± 0.01aC	0.940 ± 0.01bD	0.940 ± 0.01bD
		Chitosan Coating	0.963 ± 0.01aA	0.95 ± 0.01abAB	0.956 ± 0.01aB	0.950 ± 0.01aC	0.940 ± 0.01bCD	0.937 ± 0.01bD
		Alginate Coating	0.963 ± 0.01aA	0.96 ± 0.01aA	0.960 ± 0.01aB	0.953 ± 0.01aC	0.950 ± 0.01aD	0.942 ± 0.01aE
		Zein Coating	0.961 ± 0.01aA	0.95 ± 0.01bB	0.950 ± 0.01aC	0.950 ± 0.01aD	0.943 ± 0.01bD	0.938 ± 0.01bE
	High Dose	Control	0.970 ± 0.01aA	0.963 ± 0.01aB	0.960 ± 0.01aB	0.960 ± 0.01aB	0.957 ± 0.01bB	0.953 ± 0.01aC
		Chitosan Coating	0.962 ± 0.01aA	0.960 ± 0.01aAB	0.960 ± 0.01aAB	0.960 ± 0.01aB	0.959 ± 0.01aB	0.952 ± 0.01aC
		Alginate Coating	0.965 ± 0.01aA	0.960 ± 0.01aB	0.963 ± 0.01aB	0.960 ± 0.01aBC	0.959 ± 0.01aC	0.952 ± 0.01aD
		Zein Coating	0.964 ± 0.01aA	0.960 ± 0.01aA	0.950 ± 0.01bB	0.943 ± 0.01bC	0.941 ± 0.01cC	0.937 ± 0.01bD
DEHULLED	Low Dose	Control	0.963 ± 0.01aA	0.966 ± 0.01aA	0.953 ± 0.01aB	0.950 ± 0.01aB	0.949 ± 0.01aBC	0.944 ± 0.01aC
		Chitosan Coating	0.961 ± 0.01aA	0.960 ± 0.01abB	0.956 ± 0.01aC	0.950 ± 0.01aD	0.947 ± 0.01aE	0.941 ± 0.01aF
		Alginate Coating	0.957 ± 0.01aA	0.950 ± 0.01cB	0.950 ± 0.01aB	0.950 ± 0.01aC	0.942 ± 0.01bD	0.936 ± 0.01bE
		Zein Coating	0.965 ± 0.01aA	0.965 ± 0.01bcA	0.950 ± 0.01aAB	0.953 ± 0.01aB	0.946 ± 0.01aC	0.936 ± 0.01bD
	High Dose	Control	0.965 ± 0.01bA	0.956 ± 0.01aB	0.956 ± 0.01aB	0.950 ± 0.01aC	0.942 ± 0.01aD	0.935 ± 0.01aE
		Chitosan Coating	0.972 ± 0.01bA	0.953 ± 0.01aB	0.950 ± 0.01aC	0.943 ± 0.01aD	0.940 ± 0.01aDE	0.937 ± 0.01aE
		Alginate Coating	0.967 ± 0.01abA	0.960 ± 0.01aB	0.953 ± 0.01aC	0.950 ± 0.01aC	0.943 ± 0.01aD	0.938 ± 0.01aE
		Zein Coating	0.966 ± 0.01aA	0.953 ± 0.01aB	0.950 ± 0.01aBC	0.946 ± 0.01aCD	0.941 ± 0.01aDE	0.937 ± 0.01aE

Different small letters show significant difference in each dose of each month of storage at α = 0.05 level. Different capital letters show significant difference between the storage months each sample at α = 0.05 level.

## Data Availability

The original contributions presented in this study are included in the article. Further inquiries can be directed to the corresponding author.

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
