# Peer review of "Optimization of Fresh Storage of Pistachio (*Pistacia vera* L.) by Use of Different Coatings Under Vacuum"

_foods, 2025, doi:10.3390/foods14203533_

Round 1
Reviewer 1 Report
Comments and Suggestions for Authors
This is a comprehensive and well-designed study addressing a significant issue in the pistachio industry: extending the shelf life of the fresh product. The experimental design is robust, incorporating multiple coating types, both in-hull and dehulled samples, different inoculation levels, and a long-term storage analysis. The use of advanced statistical methods like PCA, correlation, and multi-objective optimization for data interpretation is a major strength of this manuscript. The findings are of great practical importance. To further improve the manuscript, I would like to offer the following suggestions.
- The manuscript actually involves a coating plus vacuum packaging method. Therefore, the authors need to consider adjusting the title appropriately to better reflect the research content.
- All abbreviations should be given in full before use. Abbreviations should be used throughout the rest of the text, such as FFA.
- In the Introduction and Methods sections, the authors describe three coatings (chitosan, alginate, and zein) and their corresponding antimicrobial agents (potassium sorbate, propionic acid, and EDTA). However, the formulations for the three coatings need to be explained, particularly the selection and dosage of potassium sorbate, EDTA, and propionic acid.
- In Section 3.1, it is important to clarify why sodium alginate coating was the most effective in controlling FFA accumulation in pistachios.
- Zein and chitosan coatings performed best in inhibiting the growth of Aspergillus flavus, but a multi-objective optimization analysis revealed that alginate coating was the best overall preservation solution. The authors briefly attribute this to the superiority of alginate in maintaining sensory quality. This is a key point of discussion and deserves further analysis. The authors are advised to elaborate on why sensory quality was weighted so highly in the optimization model, outweighing antibacterial efficacy. This raises a common trade-off in food preservation: pursuing ultimate microbial safety (from chitosan/zein) or better consumer acceptance (from alginate)? A thorough discussion of this trade-off would greatly enhance the article's practical relevance and academic depth.
- Section 3.4 lacks a table with statistical data on aflatoxin; please provide additional information.
- Why did zein-coated pistachios experience less water loss? The authors should provide a more in-depth analysis.
- What was the mold count on the ozone-treated pistachios in the experimental method? This needs to be quantified to ensure the rigor of the experimental protocol.
- It is recommended that some tables be converted to visualizations to help readers more clearly understand data trends or patterns.
- The authors provide a detailed description of the data changes in the Results and Analysis section and cite relevant literature, but they often lack in-depth mechanistic explanations, which is crucial for improving the quality of the manuscript.
Reviewer 2 Report
Comments and Suggestions for Authors
The article "Optimization of fresh storage of pistachio (Pistacia vera L.) by use of different coatings" describes the treatment and coating applied to fresh pistachio and influences over 5 months storage. It is a valuable study that need to address the following problems:
Abstracts should highlight the innovation of the article, as often abstract section is presented separately in search engines, it must be able to stand alone as an informative piece. In the abstract, need to focus more on the quantitative information, not qualitative one. Additionally, please avoid abbreviations in abstract and across the manuscript without fully explain them at first use (e.g. FFA).
Use uniform notation for measurement units (now for litre are used both l and L like at rows 183 and 196 but also elsewhere across the manuscript). Personally, I would recommend the use of L. Same issue with FFA and ffa (see figure 3).
Some previous work of Motelica et al (e.g. doi: 10.3390/pharmaceutics13071020) on alginate coatings for extending shelf life can help the authors to underline the importance of this study.
In figure 2, one subfigure has a black border while others do not have it. The legend in each figure is not fully shown, and makes each subfigure very clumsy. As a suggestion, use some abbreviations that should be common on all subfigures e.g. General acceptance month 1can be G1, Taste 3 Month can be T3, Texture 5 Month can be Tx5, Taste Harvest....can be T0, etc.
Subfigure d is lacking the Zein coating indication for yellow line.
Please improve the sharpness of figure 3. Sometimes the word document must be set up to not compress the images. Figure 3 is generated for all samples or for particular coating only (e.g. chitosan)? Does coating type modify the distribution on figure 3 like in figure 2?
In figure 4 some labels are overlapping, therefore affecting the legibility.
Please add main drawbacks and limitations of this study.
Conclusion section must be reworked to underline the novelty and advantages of this research, with actual numbers. The conclusion part does not highlight the salient findings and future perspective.
Reviewer 3 Report
Comments and Suggestions for Authors
The manuscript presents the effect of coating with/ without the hull of pistachio on chemical, microbiological, and sensory quality during cold storage for 5 months. The experimental design is clear, but the optimization process is not clear. More information on the optimization tool, constraints, objectives, and boundaries should be provided. Other comments are as follows.
- Why are anti-microbial compounds not included in all types of coating?
- Are the results in Table 1 from the two-way ANOVA? If so, please show the analysis of the significant effect of main parameters and their interaction. The difference between In-hull and Dehulled, Low and high doses, control, and 3 coatings can be discussed clearly.
- Line 250-251 ‘…..as they had the lowest (P<0.05) FFA.’ Is this for low or high dose?
At low-dose + in-hull, Chotosan coating and Alginate coating are not significantly different, so both can be described as the lowest FFA.
Please be careful with the explanation of the lowest and highest when the statistical analysis shows a non-significant difference or the same letters. This comment also apply to other responses (peroxide value, water activity, moisture content).
- Lines 297-300 seem to repeat Lines 292-295. Please recheck.
- Line 322, ‘…..alginate in not suitable coating materials in terms of preventing mold activity….’ Is this because of the alginate property itself, or because of no addition of antimicrobial compounds in the alginate coating? Please carefully discuss to avoid causing misunderstanding for readers.
- Table 2, how do authors calculate the statistical difference among data (<2)?
- For 3.3, 3.4, 3.5, in-hull samples show higher moisture content, higher water activity, lower mold count, and higher aflatoxin than de-hulled samples.
What does low mold count cause high aflatoxin?
- For 3.7, what is the ‘r’ value? How high is the ‘r’ to define as high, low, or no correlation? Please discuss the ‘r’, based on the defined levels.
- For 3.8, please provide the definition and how to obtain the desirability score in section 2.12 before showing the score in section 3.8.
- What is the optimum condition of preparation and coating? What are the expected responses from the optimum condition? Please add the information to section 3.8.
- Table 1 should be split into 2 tables presenting FFA and Peroxide value separately.
- Table 3 should be split into 2 tables presenting MC and water activity separately.
Reviewer 4 Report
Comments and Suggestions for Authors
Dear Authors,
your manuscript is very interesting and experimentally it is well set up. In order to improve it and increase it scientific significance, I have some suggestions:
- The Introduction is a bit long, you can shorten it if you like. More importantly, please provide clear goal of this research. One sentence would be enough at the end, like The aim of this study was to...
- Line 35: "For this reason, pistachios can cannot be consumed as fresh in other places." - please check the sentence. Please check grammatical error throughout the text.
- In the Results and Discussion section, there could be more focus on the meaning of the results and contribution of the results in relation to previous studies, rather than just describing numbers in Tables.
- Tables could be rearranged or placed on horizontally oriented paper to be more easier to read.
- The Conclusion section does not need to be that long. Just emphasize the main conclusions from the study, highlight its contribution to the industry, etc.
Best regard and good luck!
Round 2
Reviewer 2 Report
Comments and Suggestions for Authors
The authors have responded to my comments and have addressed my concerns therefore, I suggest publishing the paper titled "Optimization of fresh storage of pistachio (Pistacia vera L.) by use of different coatings under vacuum" in the current form.
Author Response
Thank you for your valuable contributions. Best regards
Reviewer 3 Report
Comments and Suggestions for Authors
From the authors’ note:
“10. What is the optimum condition of preparation and coating? What are the expected responses from the optimum condition? Please add the information to section 3.8.
Response: The optimization for coatings were not carried out. For the preparation of coatings, the literature was reviewed and the applications that yielded the best results were selected.”
To avoid confusion, please clarify.
If authors do not perform optimization, please revise sections 2.12 Statistical analyses and 3.8 Optimization.
If authors perform optimization, please specify the objectives and constraints used for the optimization. What were the FFA contents, peroxide value, total mold count, aflatoxin B1, total aflatoxin, taste, odor, texture, overall acceptability, Aw, and moisture content from the optimum preparation and coating conditions? Please add this information in Section 3.8.
